# Influenza Vaccine Effectiveness in Preventing Laboratory-Confirmed Influenza Cases and Hospitalizations in Navarre, Spain, 2022–2023

**DOI:** 10.3390/vaccines11091478

**Published:** 2023-09-12

**Authors:** Iván Martínez-Baz, Miguel Fernández-Huerta, Ana Navascués, Francisco Pozo, Camino Trobajo-Sanmartín, Itziar Casado, Aitziber Echeverria, Carmen Ezpeleta, Jesús Castilla

**Affiliations:** 1Instituto de Salud Pública de Navarra, 31003 Pamplona, Spain; imartinba@navarra.es (I.M.-B.); camino.trobajo.sanmartin@navarra.es (C.T.-S.); icasadob@navarra.es (I.C.); aitziber.echeverria.balda@navarra.es (A.E.); 2CIBER Epidemiología y Salud Pública (CIBERESP), 28029 Madrid, Spain; pacopozo@isciii.es; 3Navarre Institute for Health Research (IdiSNA), 31008 Pamplona, Spain; miguel.fernandez.huerta@navarra.es (M.F.-H.); cezpeleb@navarra.es (C.E.); 4Clinical Microbiology Department, Hospital Universitario de Navarra, 31008 Pamplona, Spain; 5National Centre for Microbiology, Instituto de Salud Carlos III, 28222 Majadahonda, Spain

**Keywords:** influenza, influenza vaccine, case-control study, vaccine effectiveness, repeated vaccination

## Abstract

We estimated influenza vaccine effectiveness (IVE) in preventing outpatient and hospitalized cases in the 2022–2023 season. A test-negative design included a representative sample of outpatients and all hospitalized patients with influenza-like illness (ILI) from October 2022 to May 2023 in Navarre, Spain. ILI patients were tested by PCR for influenza virus. Influenza vaccination status was compared between confirmed influenza cases and test-negative controls. Among 3321 ILI patients tested, IVE to prevent influenza cases was 34% (95% confidence interval (CI): 16 to 48) overall, 85% (95%CI: 63 to 94) against influenza B, and 28% (95%CI: 3 to 46) against A(H3N2). Among 558 outpatients, 222 (40%) were confirmed for influenza: 55% A(H3N2), 11% A(H1N1), and 31% B. Overall, IVE to prevent outpatient cases was 48% (95%CI: 8 to 70), 88% (95%CI: 3 to 98) against influenza B, and 50% (95%CI: −4 to 76) against A(H3N2). Of 2763 hospitalized patients, 349 (13%) were positive for influenza: 64% A(H3N2), 17% A(H1N1), and 8% B. IVE to prevent hospitalization was 24% (95%CI: −1 to 42) overall, 82% (95%CI: 49 to 93) against influenza B, and 16% (95%CI: −17 to 40) against A(H3N2). No IVE was observed in preventing influenza A(H1N1). IVE was high to prevent influenza B, moderate against A(H3N2) and null against A(H1N1). A lower proportion of influenza B cases may explain the smaller IVE in hospitalized patients than in outpatients. The null IVE against A(H1N1) was consistent with the observed antigenic drift and supports the new composition of the 2023–2024 influenza vaccine.

## 1. Introduction

The 2022–2023 influenza season was characterized by A(H3N2) influenza virus circulation until January, being displaced by influenza B/Victoria and A(H1N1) since then [1,2]. The World Health Organization recommended composition for the 2022–2023 tetravalent egg-based influenza vaccines for the Northern Hemisphere included four viruses similar to A/Victoria/2570/2019(H1N1), A/Darwin/9/2021(H3N2), B/Austria/1359417/2021 (B/Victoria lineage), and B/Phuket/3073/2013(B/Yamagata lineage) [3]. Interim estimates during the 2022–2023 influenza season from the United States, Canada, and Europe showed moderate to low influenza vaccine effectiveness (IVE) against influenza A(H3N2) and A(H1N1) viruses and higher IVE against influenza B/Victoria [4,5,6]. As these preliminary IVE estimates were obtained when influenza B and A(H1N1) viruses were still beginning to circulate [1], the final IVE estimates for the 2022–2023 season may be different [7]. In addition, these studies did not consider influenza vaccine doses administered in previous seasons, although these doses may retain some residual effect [8,9] or modify the effect of vaccination in the current season [10,11,12].

Most IVE studies evaluate only the effect in preventing primary care consultations or in preventing hospital admissions [5,6,13,14]. Studies considering both outcomes in the same area, with the same influenza vaccine products and circulating virus (sub-)types, provide a more complete view of the vaccine effect [15,16]. Furthermore, hospitalized patients and outpatients represent different groups of the same population since the first tends to be older and to present more frequently underlying conditions, and both factors could affect to the IVE [15,16].

We estimated the IVE in preventing outpatient and hospitalized cases during the 2022–2023 influenza season in Navarre, Spain.

## 2. Materials and Methods

### 2.1. Study Population and Sources of Information

The present study was performed in the region of Navarre, Spain, where IVE have been assessed on a population-based scheme since 2009 [12,16]. The Navarre Health Service provides health care to the resident population that is free of charge at the point of use. In October 2022, the tetravalent egg-produced inactivated influenza vaccine (Influvac Tetra, Mylan, Ireland) was offered free of charge to the target population, which included people aged 60 years and over, people aged 6 months and more with risk conditions, and essential workers. Other people were also able to be vaccinated if they purchased the vaccine.

Influenza surveillance relied on electronic reporting of influenza-like illness (ILI) cases by primary healthcare centers and hospitals. ILI was defined as the sudden onset of any general symptom (fever or feverishness, malaise, headache, or myalgia) in addition to any respiratory symptom (cough, sore throat, or shortness of breath) [17]. The protocol for influenza management in hospitals established early detection and swabbing of all hospitalized patients with ILI. A sentinel network of primary healthcare practices or professionals, covering a representative sample of 15% of the population, collected nasopharyngeal and pharyngeal swabs, after verbal informed consent, from their ILI patients when symptoms had appeared less than five days before. Samples were processed by real-time reverse-transcription polymerase chain reaction (RT-PCR). Influenza-type-A-positive specimens were subtyped using the HA gene as a target (Allplex, Seegene, Seoul, Republic of Korea). Positive samples with representation of each week and virus subtype were sent to the National Influenza Centre of Spain for genetic characterization based on whole sequencing of the complete hemagglutinin (HA) gene or partial sequencing of the HA1 unit, in both cases subsequently followed by phylogenetic analysis.

The variables on age, sex, presence of major chronic conditions (diabetes mellitus, heart disease, respiratory disease, cancer, immunocompromised, liver cirrhosis, stroke, dementia, rheumatic disease, and body mass index of 40 kg/m^2^ and more), and month of sample collection was obtained from the electronic records of primary healthcare. The major chronic conditions were codified according to the International Classification of Primary Care, 2nd edition [18].

Influenza vaccination status in the current and three previous seasons was obtained from the regional vaccination register [19]. Subjects were considered to be protected 14 days after vaccine administration. All patient information was linked using an individual identification number.

### 2.2. Study Design

A test-negative case-control study [20] nested in the population cohort was carried out in primary healthcare practices and hospitals in Navarre. The study included ILI cases aged 9 years and over attended in primary healthcare or hospitalized who were swabbed from October 2022 to May 2023. The Ethics Committee for Research with Medicines of Navarre approved the study protocol.

The primary analysis evaluated IVE in preventing laboratory-confirmed influenza. Cases were ILI patients attended in primary healthcare centers and hospitals and confirmed for influenza by RT-PCR. Controls were those ILI patients attended in the same settings who tested negative for influenza viruses. Healthcare workers, persons living in nursing homes, patients hospitalized for less than 24 h, and patients hospitalized before ILI symptom onset were excluded. Confirmed COVID-19 patients were also excluded from the control group since the association between influenza and COVID-19 vaccination could bias IVE estimates [21]. The IVE was also evaluated separately in primary healthcare patients and in hospital patients.

### 2.3. Statistical Analysis

Characteristics of cases and controls were compared by χ^2^ test. Crude and adjusted odds ratios (OR) with their 95% confidence intervals (CI) were calculated using logistic regression. Adjusted models included sex, age groups (9–24, 25–44, 45–64, 65–84, and ≥85 years), presence of major chronic conditions, month of sample collection, and healthcare setting (primary healthcare center or hospital) used as categorical variables. The IVE was estimated as a percentage: (1 − OR) × 100.

The main analysis considered only influenza vaccination in the current season (yes/no). An alternate analysis evaluated combinations of influenza vaccination in the current and three previous seasons in the following categories: vaccination in the current season regardless of previous seasons, vaccination in previous seasons only, and unvaccinated in the current and previous seasons as the reference category [12].

The interaction terms between the vaccination status and healthcare setting or age group were tested. Stratified analyses were carried out by age groups (9–64 and ≥65 years), for the target population for influenza vaccination, and for each virus (sub-)type (A/H1N1, A/H3N2, and B). Only months with five or more cases of a given (sub-)type were included in the analysis for that outcome.

A sensitivity analysis including COVID-19-confirmed patients in the control group was performed to evaluate the possible bias [21].

## 3. Results

### 3.1. Characteristics of Cases and Controls

A total of 3321 ILI patients were tested during the study period: 558 (17%) attended in primary healthcare centers and 2763 (83%) admitted to hospitals, of which 222 (40%) and 349 (13%) were confirmed for influenza, respectively. Of the 571 confirmed cases, 348 (61%) were due to A(H3N2), 85 (15%) to A(H1N1), 95 (17%) to influenza B, and 43 to A non-subtyped virus. Among outpatient cases, 55% were A(H3N2), 11% A(H1N1), and 31% influenza B, while in hospitalized patients, the proportion of influenza A viruses was higher: 64% A(H3N2), 17% A(H1N1), and 8% influenza B (*p* < 0.001) (Table 1 and Appendix A).

Confirmed influenza cases were more frequently younger than 65 years compared to test-negative controls (51% vs. 28%, *p* < 0.001) and had less presence of major chronic conditions (64% vs. 83%, *p* < 0.001). No statistically significant difference by sex was observed between influenza cases and test-negative controls. Among the confirmed influenza cases, 36% had received the current-season vaccine, and 44% had not been vaccinated in the current and in the three previous seasons in comparison to 56% and 25% of the test-negative controls, respectively (*p* < 0.001 for both comparisons) (Table 1).

Only 16% of influenza B cases were older than 65 years compared to 54% of influenza A(H3N2) and 60% of A(H1N1) cases (*p* < 0.001). Major chronic conditions were also more frequent in influenza A(H1N1) and A(H3N2) cases than in influenza B cases (77%, 67%, and 39%, respectively, *p* < 0.001) (Table 1).

### 3.2. Genetic Characterization

Sequence characterization of the product of amplification of the whole hemagglutinin gene or the HA1 fragment was available in 230 viruses. Of the 100 A(H3N2) strains characterized, 56 (56%) were A/Bangladesh/4005/2020-like (5% clade 2a.3 and 95% clade 2b), and 44 (44%) were A/Slovenia/8720/2022-like (77% clade 2a.1b and 23% clade 2a.1). Of the 68 A(H1N1) strains characterized, 58 (85%) were A/Sydney/5/2021-like (clade 5a.2a), and 10 (15%) were A/Norway/25089/2022-like (clade 5a.2a.1). All the 62 influenza B strains were B/Austria/1359417/2021(B/Victoria lineage)-like clade V1A.3a.2 (Table 2).

The phylogenetic trees of characterized A(H3N2) and B influenza viruses from Navarre are shown in Appendix A, respectively. The T135A substitution was present in 10% of the characterized A(H3N2) influenza viruses.

Figure 1 presents the phylogenetic tree of all the 68 A(H1N1) viruses characterized from Navarre belonging to clade 5a.2a harboring amino acid substitutions K54Q, A186T, Q189E, E224A, R259K, and K308R and to clade 5a.2a.1 harboring the same amino acid substitutions as clade 5a.2a and some others defining P137S, K142R, D260E, and T277A amino acid substitutions. The figure shows some additional amino acid substitutions in in the circulating viruses compared to the strain A/Victoria/2570/2019 included in the egg-based vaccine recommended for the 2022–2023 northern hemisphere influenza season and fewer differences with the vaccine virus A/Victoria/4897/2022 (belonging to the 5a.2a.1 clade) recommended for the egg-based northern hemisphere 2023–2024 season.

### 3.3. Influenza Vaccine Effectiveness Estimates

The overall adjusted IVE estimate was 34% (95% CI: 16 to 48) when previous vaccination was not considered and was similar (36%, 95% CI: 16 to 51) when individuals vaccinated in any of the three previous seasons were excluded from the reference category. Individuals unvaccinated in the current season did not show a significant residual effect of previous doses (6%, 95% CI: −27 to 30) (Figure 2 and Appendix A).

Stratified analysis by age groups (9–64 and ≥65 years) found similar IVE estimates for current-season vaccination (31% and 28%, respectively, *p-_interaction_* = 0.920). The estimate was also similar for the target population for influenza vaccination (35%). The IVE estimates were 85% (95% CI: 63 to 94) and 28% (95% CI: 3 to 46) in preventing influenza B and A(H3N2) cases, respectively. No effect was observed in preventing influenza A(H1N1) cases (−37%; 95% CI: −159 to 27) (Figure 2).

IVE was higher in preventing outpatient cases than hospitalized cases: 48% (95% CI: 8 to 70) vs. 24% (95% CI: −1 to 42), *p-_interaction_* = 0.028. No differences in IVE were observed between people aged 9 to 64 and those aged 65 years and older, in both primary healthcare (45% and 48%, *p-_interaction_* = 0.777) and hospital settings (12% and 25%, *p-_interaction_* = 0.585). Similar estimates were observed in analyses restricted to the target population for influenza vaccination. In both settings, no effect was observed in individuals unvaccinated in the current season who had been vaccinated in any previous season (Table 3 and Table 4 and Appendix A).

The IVE estimates in preventing influenza outpatient cases were 50% (95% CI: −4 to 76), 7% (95% CI: −233 to 72), and 88% (95% CI: 3 to 98) against influenza A(H3N2), A(H1N1), and B, respectively, and to prevent hospitalizations were 16% (95% CI: −17 to 40), −53% (95% CI: −233 to 30), and 82% (95% CI: 49 to 93), respectively (Table 3 and Table 4).

The sensitivity analysis including the COVID-19-positive patients in the control group provided similar IVE estimates (Appendix A).

## 4. Discussion

The present study found a moderate IVE, on average, during the 2022–2023 influenza season in Navarre, Spain, although estimates against each specific circulating virus (sub-) type were very diverse. This influenza season was characterized by circulation of three (sub-)types of influenza viruses in different moments. The IVE was high in preventing influenza B cases (85%), moderate to low against influenza A(H3N2) (28%), and null against influenza A(H1N1). All the 62 influenza B strains were B/Austria/1359417/2021(B/Victoria lineage)-like, which was similar to the B/Victoria component included in the 2022–2023 influenza vaccine. While the A(H3N2) strain included in the vaccine (A/Darwin/9/2021) inhibited the clade 2a1b.2a.2 represented by Bangladesh-like viruses well [3], four different clades were detected among the genetically characterized viruses, some of them with substantial divergence compared with the vaccine component [2]. The A(H1N1) component included in the vaccine recognized 5a.2 viruses well but, in our study, presented a null IVE against the circulating A/Sydney/5/2021-like (clade 5a.2a) and A/Norway/25089/2022-like (clade 5a.2a.1) strains, which were closest to the strain recommended for the 2023–2024 vaccine [2,3,22].

The IVE in preventing outpatient cases was moderate (48%), and the effect was similar in patients aged 9 to 64 years (45%) and 65 years and more (48%). These IVE estimates were consistent with interim results from a European Network, the United States, and Canada (40–54%) [4,5,6]. The present study found a low IVE in preventing hospitalized influenza cases (24%), which was lower compared to the interim estimate obtained in a European study (50%) [4] and the end-of-season estimates in an Italian hospital (57%) [23]. This finding may be explained in part because, compared to outpatients, for in-hospital patients in Navarre, the proportion of influenza A(H3N2) and A(H1N1) cases was higher, which included influenza subtypes with low IVE.

The T135A substitution was only present in 10% of the characterized A(H3N2) influenza viruses from Navarre. Variation in position 135 has been associated with IVE [5].

The null IVE observed in Navarre against A(H1N1) virus contrasted with an appreciable protection observed in other studies [4,5,6,21]. This finding suggests the circulation of strains with more pronounced antigenic drift in Navarre and supports the change in the A(H1N1) vaccine component recommended by the World Health Organization for the influenza vaccines for use in the northern hemisphere for the 2023–2024 influenza season [22]. According to this recommendation, recently circulating viruses would better match to the vaccine virus A/Victoria/4897/2022 (belonging to the 5a.2a.1 clade) than the previous strain A/Victoria/2570/2019 included in the egg-based vaccine recommended for the 2022–2023 season [22].

In the present study, the overall IVE estimate to prevent outpatient cases (48%) was higher in comparison to preventing inpatient cases (24%). However, no differences in the IVE estimates were observed between settings in the analysis stratified by virus (sub-) types. These findings may be explained by the lower proportion of influenza B cases in hospitalized patients than in outpatients and the high vaccine effectiveness found to prevent cases of this type of virus. A previous study described, in some seasons, similar IVE estimates in preventing outpatient cases and hospitalizations with confirmed influenza but, in other seasons, an additional effect of the influenza vaccination in reducing the risk of hospital admission among people in whom the vaccine failed to prevent influenza infection [16].

The strengths of the present study include the test-negative design that is considered the recommended design for IVE assessment [20,24,25,26,27]. All cases were laboratory-confirmed for influenza patients, and controls were test-negative individuals. Patients and healthcare professionals were blinded to the case or control status, which improved the comparability and reduced the possibility of selection bias. Subjects were recruited from the same region, received the same vaccine brand, and were exposed to common circulating viruses. Inclusion of primary healthcare centers and hospital settings provides complementary insights of the IVE in the same population. To avoid information bias [12], the vaccination status was obtained from the regional vaccination registry [19].

In contrast with findings from previous influenza seasons [9,12], the inclusion of vaccines received in the three previous seasons did not appreciably change the estimates of the current-season IVE. No effect was observed in individuals unvaccinated in the current season who had been vaccinated in any previous season, nor did vaccination history change the IVE estimate of the current-season vaccination.

The inclusion of COVID-19-positive patients in the control group did not appreciably modify the results, which rules out the possible bias suggested by other authors [21].

Some limitations may affect this study. This study was carried out in a region where vaccination was recommended for persons aged 60 years and older and those with risk conditions, and only a tetravalent egg-produced inactivated vaccine was used during the study period; therefore, caution should be taken in making generalizations of our results to other countries with different influenza vaccination indications, vaccination coverage, or where other vaccines brands were used. As the statistical power was reduced in some analyses, the estimates of IVE in the elderly population attended in primary healthcare or in the hospitalized young adults should be interpreted cautiously. This study included individuals with different chances for influenza vaccination due to their age or presence of comorbidities; therefore, analyses were adjusted for these variables to control for potential confounders [28,29].

## 5. Conclusions

In conclusion, the vaccine effectiveness during the 2022–2023 influenza season in Navarre was moderate on average, high to prevent influenza B, moderate to low against influenza A(H3N2), and null against influenza A(H1N1). A lower proportion of influenza B cases in hospitalized patients explains the lower IVE in preventing hospitalized influenza cases than outpatient cases. These results support the recommended composition of influenza vaccines to be used in the 2023–2024 influenza season and reinforce the annual recommendation of influenza vaccination.

## Figures and Tables

**Figure 1 vaccines-11-01478-f001:**
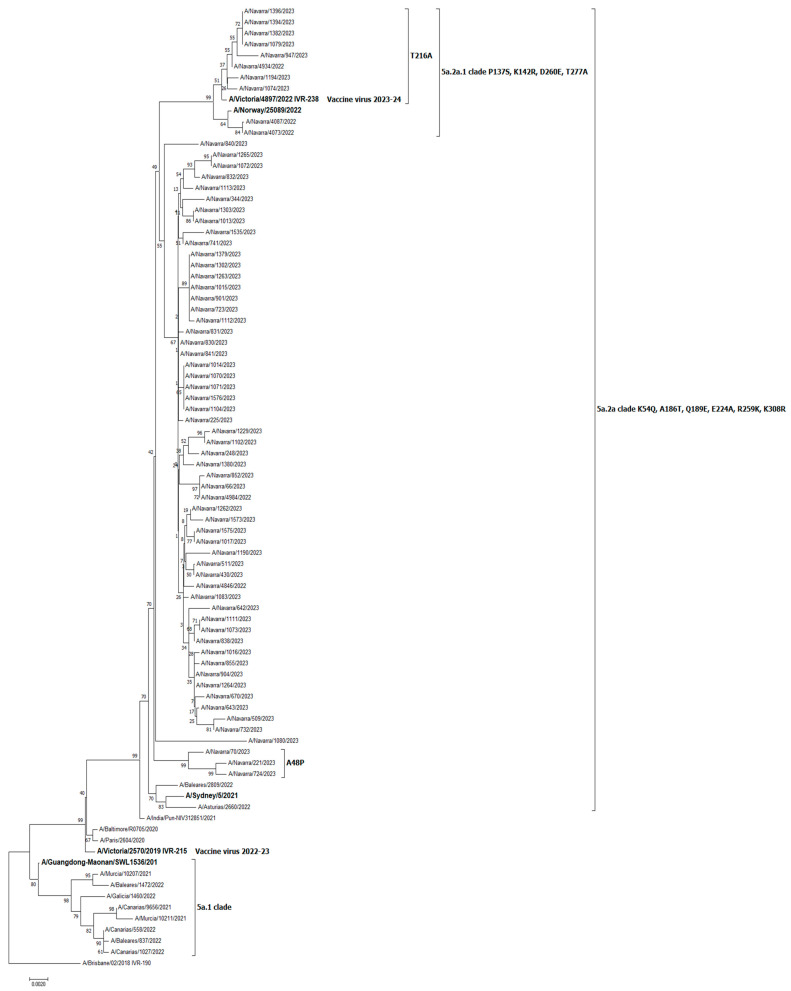
Phylogenetic tree representing all characterized A(H1N1) influenza viruses from Navarre, Spain.

**Figure 2 vaccines-11-01478-f002:**
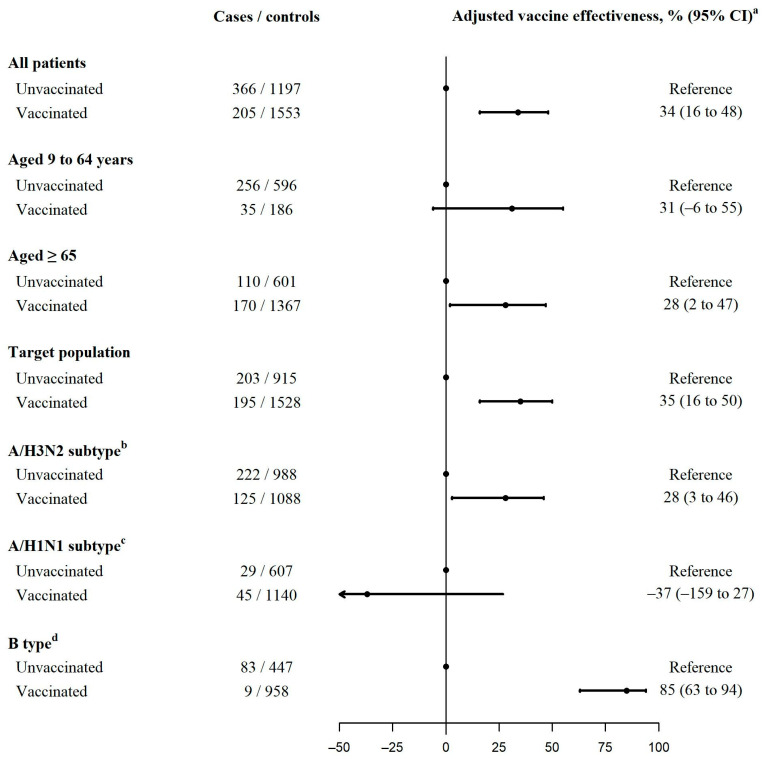
Influenza vaccine effectiveness in preventing laboratory-confirmed influenza. Pooled analysis of primary healthcare patients and hospitalized patients. ^a^ Vaccine effectiveness adjusted by sex, age groups, presence of major chronic conditions, healthcare setting (primary healthcare or hospital), and month of sample collection. ^b^ The analysis of A/H3N2 covered the period October 2022 to February 2023. ^c^ The analysis of A/H1N1 covered the period December 2022 to March 2023. ^d^ The analysis of B covered the period January to April 2023.

**Table 1 vaccines-11-01478-t001:** Baseline characteristics of individuals who consulted or were hospitalized for influenza-like illness according to the result of the influenza test.

Characteristics	Negative Controls *n* (%) *n* = 2750	All Confirmed Influenza Cases, *n* (%) *n* = 571	*p*-Value	Influenza A(H1N1) *n* (%) *n* = 85	Influenza A(H3N2) *n* (%) *n* = 348	Influenza B *n* (%) *n* = 95	*p*-Value
**Age group in years**			<0.001				<0.001
9–24	90 (3)	58 (10)		2 (2)	35 (10)	18 (19)	
25–44	191 (7)	114 (20)		11 (13)	54 (16)	48 (51)	
45–64	501 (18)	119 (21)		21 (25)	71 (20)	13 (14)	
65–84	1238 (45)	201 (35)		41 (48)	130 (37)	7 (7)	
≥85	730 (27)	79 (14)		10 (12)	58 (17)	9 (9)	
**Sex**			0.614				0.002
Male	1390 (50)	282 (49)		42 (49)	179 (51)	39 (41)	
Female	1360 (50)	289 (51)		43 (51)	169 (49)	56 (59)	
**Presence of major chronic conditions**			<0.001				<0.001
No	473 (17)	206 (36)		20 (23)	116 (33)	58 (61)	
Yes	2277 (83)	365 (64)		65 (77)	232 (67)	37 (39)	
**Vaccination in the current season**			<0.001				<0.001
No	1197 (43)	366 (64)		34 (40)	222 (64)	85 (89)	
Yes	1553 (57)	205 (36)		51 (60)	126 (36)	10 (11)	
**Vaccination in the current and three prior seasons**			<0.001				<0.001
Never vaccinated	678 (25)	252 (44)		27 (32)	131 (38)	77 (81)	
Vaccination in prior seasons only	519 (19)	114 (20)		7 (8)	91 (26)	8 (8)	
Current season vaccination	153 (56)	205 (36)		51 (60)	126 (36)	10 (11)	
**Healthcare setting**			<0.001				<0.001
Primary healthcare center	336 (12)	222 (39)		24 (28)	123 (35)	68 (72)	
Hospital	2414 (88)	349 (61)		61 (72)	225 (65)	27 (28)	
**Month of sample collection**			<0.001				<0.001
October 2022	315 (12)	47 (8)		3 (4)	42 (12)	0 (0)	
November 2022	335 (12)	104 (18)		3 (4)	95 (27)	0 (0)	
December 2022	591 (22)	164 (29)		9 (11)	148 (43)	2 (2)	
January 2023	541 (20)	86 (15)		26 (31)	50 (14)	18 (19)	
February 2023	294 (11)	63 (11)		36 (42)	12 (3)	21 (22)	
March 2023	321 (12)	91 (16)		8 (9)	1 (0.3)	47 (50)	
April 2023	249 (9)	15 (3)		0 (0)	0 (0)	6 (6)	
May 2023	104 (4)	1 (0.2)		0 (0)	0 (0)	1 (1)	

**Table 2 vaccines-11-01478-t002:** Influenza viruses characterized by clade.

Genetic Virus Characterization	Clade	*n* (%)
**Influenza A(H1N1)pdm09 (*n* = 68)**		
A/Sydney/5/2021 (*n* = 58)	5a.2a	58 (85)
A/Norway/25089/2022 (*n* = 10)	5a.2a.1	10 (15)
**Influenza A(H3N2) (*n* = 100)**		
A/Bangladesh/4005/2020 (*n* = 56)	2a.3 2b	3 (5) 53 (95)
A/Slovenia/8720/2022 (*n* = 44)	2a.1b 2a.1	34 (77) 10 (23)
**Influenza B/Victoria (*n* = 62)**		
B/Austria/1359417/2021	V1A.3a.2	62 (100)

**Table 3 vaccines-11-01478-t003:** Influenza vaccine effectiveness in preventing outpatient cases with laboratory-confirmed influenza.

	Cases/Controls	Crude Vaccine Effectiveness, % (95% CI)	Adjusted Vaccine Effectiveness, % (95% CI) ^a^	*p*-Value
**All outpatients**				
Unvaccinated	196/238	1	1	
Vaccinated	26/98	68 (48 to 80)	48 (8 to 70)	0.026
**Aged 9 to 64 years**				
Unvaccinated	189/216	1	1	
Vaccinated	15/39	56 (18 to 76)	45 (−7 to 72)	0.077
**Aged ≥65 years**				
Unvaccinated	7/22	1	1	
Vaccinated	11/59	41 (−70 to 80)	48 (−83 to 85)	0.312
**Target population**				
Unvaccinated	58/93	1	1	
Vaccinated	22/86	59 (27 to 77)	47 (−8 to 73)	0.080
**A/H3N2 subtype ^b^**				
Unvaccinated	107/183	1	1	
Vaccinated	15/77	67 (39 to 82)	50 (−4 to 76)	0.063
**A/H1N1 subtype ^c^**				
Unvaccinated	17/164	1	1	
Vaccinated	6/80	28 (−91 to 72)	7 (−233 to 74)	0.908
**B type ^d^**				
Unvaccinated	66/116	1	1	
Vaccinated	1/53	97 (75 to 99)	88 (3 to 98)	0.046

Abbreviations: CI, confidence interval. ^a^ Vaccine effectiveness adjusted by sex, age groups (9–24, 25–44, 45–64, 65–84, and ≥85 years), presence of major chronic conditions, and month of sample collection. ^b^ The analysis of A/H3N2 covered the period October 2022 to February 2023. ^c^ The analysis of A/H1N1 covered the period December 2022 to March 2023. ^d^ The analysis of B covered the period January to April 2023.

**Table 4 vaccines-11-01478-t004:** Influenza vaccine effectiveness in preventing hospitalized patients with laboratory-confirmed influenza.

	Cases/Controls	Crude Vaccine Effectiveness, % (95% CI)	Adjusted Vaccine Effectiveness, % (95% CI) ^a^	*p*-Value
**All inpatients**				
Unvaccinated	170/959	1	1	
Vaccinated	179/1455	31 (13 to 45)	24 (−1 to 42)	0.055
**Aged 9 to 64 years**				
Unvaccinated	67/380	1	1	
Vaccinated	20/147	23 (−32 to 55)	12 (−57 to 50)	0.671
**Aged ≥65 years**				
Unvaccinated	103/579	1	1	
Vaccinated	159/1308	32 (11 to 48)	25 (−2 to 46)	0.067
**Target population**				
Unvaccinated	145/822	1	1	
Vaccinated	173/1442	32 (14 to 46)	31 (9 to 48)	0.009
**A/H3N2 subtype ^b^**				
Unvaccinated	115/780	1	1	
Vaccinated	110/998	25 (1 to 43)	16 (−17 to 40)	0.301
**A/H1N1 subtype ^c^**				
Unvaccinated	12/443	1	1	
Vaccinated	39/1060	−35 (−162 to 29)	−53 (−233 to 30)	0.284
**B type ^d^**				
Unvaccinated	17/367	1	1	
Vaccinated	9/973	80 (55 to 91)	82 (49 to 93)	0.001

Abbreviations: CI, confidence interval. ^a^ Vaccine effectiveness adjusted by sex, age groups (9–24, 25–44, 45–64, 65–84, and ≥85 years), presence of major chronic conditions, and month of sample collection. ^b^ The analysis of A/H3N2 covered the period October 2022 to February 2023. ^c^ The analysis of A/H1N1 covered the period December 2022 to March 2023. ^d^ The analysis of B covered the period January to April 2023.

## Data Availability

The data presented in this study are available on request from the corresponding author. The data are not publicly available due to privacy issues.

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
