# Peer review of "Influenza Vaccine Effectiveness in Preventing Laboratory-Confirmed Influenza Cases and Hospitalizations in Navarre, Spain, 2022–2023"

_vaccines, 2023, doi:10.3390/vaccines11091478_

Round 1

Reviewer 1 Report

In the presented manuscript, it is reported the case of influenza A(H3N2) until January and influenza B and A(H1N1) since then, during 2022-2023 influenza season in a region of Spain. The authors estimated the influenza vaccine effectiveness (IVE) in preventing outpatient and hospitalized cases in the 2022-2023 season. Using molecular methods and advanced static methods, they also compared the influenza vaccination status between confirmed influenza cases and test-negative controls. The methodology seems really appropriate and the manuscript is written properly scientifically, but I would suggest to the authors substitute some of the presented tables with relative graphics. In this way, it would be even easier to discuss the relative outputs with the results of other authors in similar publications and it would be even easier for the reader. 

In addition, I would please to also extend the Conclusion part by including other cases if possible in Spain previous to go directly to support the recommendations at the international level.

Author Response

Thank you very much for your comments.

We have added two figures. Figure 2 (page 7) presents the vaccine effectiveness estimates according to the reviewer's recommendation.

There are currently no published results on the influenza vaccine effectiveness from other sites in Spain referred to the 2022-2023 season. However, our results are similar to those found in the interim study in Europe, as we say in the discussion.

Reviewer 2 Report

Reviewer’s comment-

The manuscript entitled “Influenza vaccine effectiveness in preventing laboratory confirmed influenza cases and hospitalizations in Navarre, Spain, 2022-2023” by Ivan Bar et al. The study mentioned the Influenza vaccine's effectiveness in preventing outpatients and hospitalized patients and Influenza-like illness in the region of Navarre, Spain for the year 2022-23. The study represents the moderate effect of the Influenza vaccine (Influvac, Tetra) in the 2022-23 season, the vaccine effectiveness was high for Influenza B, low for H1N1, and null for H3N2 strains of the Influenza virus. However, the manuscript lacks at several points with few limitations of the study, some of them were already mentioned by the authors.

Minor comments:

1.     The entire manuscript should be revised for minor grammatical errors and some parts in the results section such as lines number 140 to 146 on Page No: 3 should be rewritten in order to bring clarity to the sentence. 

2.     The variables used in the study to evaluate the effectiveness of Influenza vaccination were Nominal or Ordinal.

3.     The authors are advised to check the interaction P-value among the non-vaccinated, pre-vaccinated, and revaccinated groups to determine if there is an interaction between the treatment effect and the considered variables that define subgroups.

4.     In the materials and methods section, it has been mentioned that in October 2022, the tetravalent egg produced inactivated vaccine (Influvac tetra) which includes the virus composition for the 2023-24 and recommended by WHO (northern hemisphere), was given to the target population, aged 60 years and the chronic illness patient, the limitation is that the vaccine should be given in adults and younger population to see the effectiveness of the vaccine in these population.

5.     In the study design the total number of patients for Influenza-like illness (ILI) and Influenza vaccine effectiveness (IVI) with the age groups should be mentioned.

6.     The positive number of influenza cases was first determined by the real-time PCR, there were H1N1, H3N2, and Influenza B viruses were checked by RT-PCR, Please mention which gene was used for the real-time PCR study.

7.     In the materials and methods section 2.1 and result section 3.2, it was mentioned that the positive samples were selected for the whole sequencing of the HA and HA1 unit, and the summarized information for the clades was given in the supplementary table S2, please provide phylogeny for the same in the form of figure for the better interpretation of the results. 

Minor editing is needed for English grammar and language.

Author Response

Thank you very much for your comments. We respond your comments point by point.

1.We have revised the entire manuscript, and rewritten the lines that you mention.

2. All variables were used as categorical. Now we say in the methods section, lines 116-119: “Adjusted models included sex, age groups (9–24, 25–44, 45–64, 65–84, and ≥85 years), presence of major chronic conditions, month of sample collection, and healthcare setting (primary healthcare or hospital) used as categorical variables.”

3. In an alternate analysis (supplementary table S2), we evaluated the influenza vaccine effectiveness combining the vaccination in the current and three previous seasons. In that analysis, we rule out a residual effect of vaccines received in previous seasons. For this reason, we have included the main analysis considering only vaccination in the current season.

We checked the interaction terms between the vaccination status (in two and three categories) and healthcare setting and age group. The results were similar regardless of the inclusion or not of vaccination in previous seasons in the categories of vaccination status.

4. We have rewrite in methods: “the target population, which included people aged 60 years and over, people aged 6 months or more with major chronic conditions, and essential workers. Other people were also able to be vaccinated if they purchased the vaccine”

For this reason, we performed the analyses with all the individuals who met the inclusion criteria. In addition, we also carried out the specific analysis in the target population for influenza vaccination, obtaining similar results to overall analysis.

5. All hospitalized patients with ILI and RT-PCR test for influenza are included in the study (n=2763, see line 135). In the methods section we say: "A sentinel network of primary healthcare professionals, covering a representative sample of 16% of the population" and in the results section: “A total of 3321 ILI patients were tested during the study period, 558 (17%) attended in primary healthcare centres and 2763 (83%) admitted to hospitals”.

6. We have added (lines 80-83): “Samples were processed by real-time reverse-transcription polymerase chain reaction (RT-PCR) (Allplex SARS-Cov-2/FluA/FluB/RSV Assay, Seegene, Seoul, South Korea). Influenza type A positive specimens were subtyped using the HA gene as a target (Allplex Respiratory Panel 1, Flu/RSV/FluA subtyping, Seegene, Seoul, South Korea).”

7. We have added included in the main manuscript table 2 and figure 1 with the phylogenetic tree of the A(H1N1) virus. The trees for the A(H3N2) and B viruses are included in the supplementary material.

English grammar has been revised.

Reviewer 3 Report

The authors did good work trying to define the IVE for the inflenza vaccine for Navarre, Spain for 2022-2023. The authors had good experimental design with emphasis on controls. The authors control the type of vaccine and negative control patients which made this study robust.

The authors will have to address the issue of patient consent in this study.

However, as the authors pointed out in their discussion their study is limited. The limited nature of scope of this study is observed in it being limited to 558 patients form Navarre, Spain and as noted by the authors the conclusions noted here can’t be used in a different region or generalized. The impact of this study beyond this region is limited and unknown. There were no significantly different findings from papers published previously on IVE’s of influeza vaccine, some of which are cited by the authors.

Similarly, generalized statements were made from limited sample size “vaccines received from three previous seasons did not appreciably change the estimates of current-season IVE”. This was different from other regions and brings us to the significance of this study in enhancing our knowledge or improving the outcome. 

Author Response

AUTHORS’ RESPONSE: Thank you very much for your comments.

The authors will have to address the issue of patient consent in this study.

AUTHORS’ RESPONSE: In the methods we say (line 78): “…collected nasopharyngeal and pharyngeal swabs, after verbal informed consent,…”. Besides, in the Institutional Review Board Statement section we say: ”This study was approved by the Navarre’s Ethical Committee for Clinical Research (PI2020/45), which waived the requirement of obtaining informed consent.”

However, as the authors pointed out in their discussion their study is limited. The limited nature of scope of this study is observed in it being limited to 558 patients form Navarre, Spain and as noted by the authors the conclusions noted here can’t be used in a different region or generalized. The impact of this study beyond this region is limited and unknown. There were no significantly different findings from papers published previously on IVE’s of influeza vaccine, some of which are cited by the authors.

AUTHORS’ RESPONSE: Our study included 3321 ILI patients. Therefore, only in some stratified analyses the statistical power was reduced. Our estimates of influenza vaccine effectiveness in preventing hospitalizations was lower compared to the interim estimates obtained in the European study or in the end-of-season in an Italian hospital, as we comment in the discussion. This may be explained in part, due to different subtypes of influenza virus circulation.

We do not stated that our results cannot be used for other regions. Our sentence is only for caution in this use. We say: “..caution should be taken in making generalizations of our results to other countries with different influenza vaccination indications, vaccination coverage or where other types of vaccines were used.”

Although most of our results are consistent with interim reports, we found null vaccine effectiveness against influenza A(H1N1), which is a different result compared to other studies. Lines 284-285: “The null IVE observed in Navarre against A(H1N1) virus contrasted with an appreciable protection observed in other studies [4-6,22]”

Similarly, generalized statements were made from limited sample size “vaccines received from three previous seasons did not appreciably change the estimates of current-season IVE”. This was different from other regions and brings us to the significance of this study in enhancing our knowledge or improving the outcome. 

AUTHORS’ RESPONSE: Thanks for your comment. We modify the sentence. However, to our knowledge, no other study site has yet evaluated the effect of vaccine doses administered in previous seasons on the 2022-2023 season. We say in the discussion (lines 312-316: “In contrast with findings from previous influenza seasons [8,11], the inclusion of vaccines received in the three previous seasons did not appreciably change the estimates of the current-season IVE. No effect was observed in individuals unvaccinated in the current season who had been vaccinated in any previous season, nor did vaccination history change the IVE estimate of the current-season vaccination.”

Round 2

Reviewer 3 Report

The authors have adressed most comments except the issue of this study being limited to the region. The authors thorugh thier language adressed generalized statements and hence can be approved for publication.